# Genetic Variants in the NOD-like Receptor Signaling Pathway Are Associated with HIV-1/AIDS in a Northern Chinese Population

**DOI:** 10.3390/ijms26083484

**Published:** 2025-04-08

**Authors:** Tingyu Pan, Yi Yang, Xia Zhang, Chenghong You, Jiawei Wu, Lidan Xu, Wei Ji, Xueyuan Jia, Jie Wu, Wenjing Sun, Songbin Fu, Xuelong Zhang, Yuandong Qiao

**Affiliations:** 1Department of Medical Genetics, School of Basic Medical Sciences, Harbin Medical University, Harbin 150081, China; 15034934154@163.com (T.P.); youchenghong11@163.com (C.Y.); xuld@ems.hrbmu.edu.cn (L.X.); jiwei@ems.hrbmu.edu.cn (W.J.); wujie@hrbmu.edu.cn (J.W.);; 2Key Laboratory of Frigid Zone Exercise Health Research and Translation in Heilongjiang Province, Harbin Medical University, Daqing 163319, China; 3Key Laboratory of Preservation of Human Genetic Resources and Disease Control in China, Harbin Medical University, Ministry of Education, Harbin 150081, China

**Keywords:** HIV-1, AIDS, NOD-like receptor, innate immune, SNP

## Abstract

The NOD-like receptor (NLR) signaling pathway may influence human immunodeficiency virus (HIV) clearance and CD4^+^ T cell recovery through inflammatory responses, but its specific mechanism requires further investigation. A deeper understanding of genetic variations can provide new insights into the biological mechanisms underlying the occurrence and development of immunodeficiency syndrome (AIDS). By utilizing multiple bioinformatic analyses and functional annotations, we identified single-nucleotide polymorphisms (SNPs) in the NLR signaling pathway that may affect HIV-1 infection and AIDS progression. Then, a case–control study was performed to screen risk-related variants by genotyping candidate SNPs in a sample of 500 men who have sex with men (MSM) with HIV-1 and 500 healthy controls from the Han population in Northern China. The results revealed significant association between five SNPs (*NLRP3 rs4612666*, *MAVS rs17857295*, *MAVS rs6084497*, *MAVS rs16989000*, and *JAK1 rs4244165*) and HIV-1 infection. Interestingly, the gene–gene interaction model composed of five SNPs exhibited a cumulative effect on the disease. Specially, the increase in risk alleles carried by the samples elevated the risk of contracting HIV-1. In addition, three SNPs (*IL1B rs1143623*, *STAT1 rs1467199* and *STAT1 rs2066804*) were associated with CD4^+^ T cell counts in patients with AIDS. Three SNPs (*OAS1 rs1131454*, *NLRP3 rs10754558*, and *MAVS rs867335*) were found to be related to the clinical staging of AIDS. This finding provides insights into the genetic variants in NLR signaling pathway genes in HIV-1 infection and AIDS progression among MSM in Northern China.

## 1. Introduction

Human immunodeficiency virus (HIV) is a single-stranded retrovirus that invades host cells and reverse transcribes itself into host cell DNA, leading to acquired immunodeficiency syndrome (AIDS) [1]. In 2022, about 29.8 million of the 39 million people living with HIV (PLWH) had received treatment [2]. In 2023, there were approximately 6.7 million PLWH in the Asia and Pacific region. Compared to 2010, the number of new infections has decreased by 13%, and the number of AIDS-related deaths has declined by 51%, but new HIV infections among men who have sex with men (MSM) have increased by 32% [3]. Greater attention should be given to key susceptible populations. The decrease in mortality rates is closely associated with the widespread use of long-term antiretroviral therapy (ART). However, despite the effectiveness of ART, it cannot completely eliminate challenges such as viral reservoirs, poor patient compliance, and drug resistance [4]. Therefore, further research and optimization are still needed. Recent studies show that innate immunity will limit viral growth and slow CD4^+^ T cell depletion when ART is stopped. Innate immune cells, such as natural killer cells and dendritic cells, may play a role in reversing viral latency [5,6].

The innate immune system serves as the first line of defense against pathogens, utilizing pattern recognition receptors (PRRs) to detect invading pathogens [7]. PRRs are classified into membrane-bound and cytosolic-binding types, with NOD-like receptors (NLRs) being a type of cytosolic-binding PRRs [8]. The activation of the NLRP3 inflammasome in HIV-1 infection is caused by multiple mechanisms, including potassium efflux, mitochondrial release of reactive oxygen species, and lysosomal damage [9]. The assembly of the inflammasome is a core step in the activation of the NLR signaling pathway. The NLRP3 inflammasome mediates Caspase-1 to convert precursors IL-1β/IL-18 into their active forms, while concurrently activating GSDMD to form membrane pores which facilitate pyroptosis and the release of inflammatory factors, thereby controlling the viral infection [10]. The NLR signaling pathway plays a dual role in the regulation of HIV-1 infection. On one hand, it can modulate the inflammatory response and immune reaction during HIV-1 infection [11]. Under the stimulation of HIV-1, NLRs can trigger an inflammatory response, prompting the body to eliminate pathogens [12]. On the other hand, the activation of the NLRP3 inflammasome in patients with chronic AIDS maintains the body in a state of chronic inflammation, which adversely affects the recovery of CD4^+^ T cells during ART treatment and promotes the occurrence of complications, correlating with a poorer prognosis [13,14]. Achieving precise regulation remains a critical challenge, as over-suppression can compromise patients’ immune capability, while over-activation may lead to immune exhaustion, ultimately reducing patients’ immunity. The specific mechanisms of these impacts necessitate further exploration and discussion.

The relationship between genetic variations in NLR signaling pathway-related genes and various complex diseases has been widely studied. For instance, *NLRP3* polymorphisms were reported to affect inflammatory factor concentrations in abdominal aortic aneurysm and were significantly associated with the risk of gouty arthritis [15,16]. Additionally, splice variants of *OAS1* were considered to have a protective effect on the severity of COVID-19 [17]. However, there are few reports on the association between the genetic polymorphisms of NLR signaling pathway genes and HIV-1 infection and AIDS progression. In this study, bioinformatic analysis indicated a potential relationship between the NLR signaling pathway and AIDS. Based on this finding, we proceeded to conduct an association study to investigate whether genetic variations in key genes within this regulatory axis influenced the disease. The aim of this study was to explore the effect of genetic variations on HIV-1 infection, the regulation of inflammatory response, and the progression of AIDS.

## 2. Results

### 2.1. Selection of NOD-like Receptor Signaling Pathway

The overall bioinformatic research process is shown in Figure 1. Initially, three datasets (GSE157198, GSE29429, and GSE2171) related to HIV-1 infection were selected from the Gene Expression Omnibus (GEO) database. Differential gene expression analysis was conducted on healthy controls and patients with AIDS not undergoing ART in three datasets (Figure 2A and Appendix A). Subsequently, gene set enrichment analysis (GSEA) was conducted on the differential gene expression results for each dataset. The enrichment results showed that, in addition to basic life pathways such as “DNA replication” and “cell cycle”, immune-related signaling pathways such as “T cell receptor signaling pathway” and “IL-17 signaling pathway” were significantly enriched (Figure 2B and Appendix A). In order to comprehensively explore the influence of differentially expressed genes (DEGs) on the pathway, we identified genes that appeared in at least two out of three datasets as gene set 1 and conducted Kyoto Encyclopedia of Genes and Genomes (KEGG) pathway enrichment on them (Figure 2C). The results indicated significant enrichment in the “NOD-like receptor signaling pathway” and “Measles” (Figure 2D).

Two datasets, GSE29429 and GSE195434, containing clinical information such as gender, CD4^+^ T cell count, and viral load, were selected from the GEO database for further analysis. Deconvolution analysis using CIBERSORTx in GSE29429 revealed significant differences in the composition of different CD4^+^ T cell subtypes between AIDS and healthy samples (Figure 3A). Subsequently, weighted correlation network analysis (WGCNA) was employed to identify gene modules significantly associated with clinical information across the two datasets under optimal threshold conditions (*p* < 0.05) (Figure 3B,D). The “MEyellow” gene module in GSE29429 contained 219 genes such as *IFITM1*, *NOD1*, and *OAS1*. The KEGG pathway enrichment analysis indicated that the “MEyellow” gene module was enriched in the NOD-like receptor signaling pathway (Figure 3C). The GSE195434 dataset included samples from multiple couples in whom one partner was infected with HIV-1 while the other was healthy. By recording the time at which the healthy partner became infected and using WGCNA to identify gene modules related to these time changes, we screened for genes with significant expression pattern alterations during HIV-1 infection and disease progression (Figure 3D). The “MEmagenta” gene module contained 38 genes such as *STAT1*, *STAT2*, and *OAS1*. The enrichment analysis also showed its involvement in the NOD-like receptor signaling pathway (Figure 3E). These findings suggested that genes in the NLR signaling pathway were associated with clinical indicators such as CD4^+^ T cell count, viral load, and infection time in individuals with HIV-1.

Based on the above bioinformatic analyses, we identified the intersection of module genes from two WGCNA analyses, which we referred to as gene set 2, and conducted KEGG pathway enrichment on them (Appendix A). Then, we intersected gene set 1 and gene set 2 and performed a KEGG pathway enrichment analysis on each resulting gene (Appendix A). The enrichment results indicated significant involvement in the “NOD-like receptor signaling pathway” and “Necroptosis”. Therefore, based on the bioinformatic findings of this study and guidance from reports in the literature, we focused our research on the NOD-like receptor signaling pathway. We then compiled a target gene set for our genetic association study by merging genes enriched in this pathway with genes related to AIDS reported in the literature. This target gene set included *CASP1*, *STAT1*, *OAS1*, *IL18*, *GSDMD*, *NLRP3*, *IL1B*, *MAVS*, *JAK1*, *STAT2*, and *PYCARD*.

### 2.2. Selection and Annotation of Functional SNPs

Single-nucleotide polymorphisms (SNPs) on target genes with a minor allele frequency (MAF) greater than 0.05 were selected from the 1000 Genomes Project data. Functional annotation was then performed on 3259 SNPs across 11 genes using multiple functional prediction databases. After performing linkage disequilibrium analysis (r^2^ > 0.8), 37 functional SNPs distributed on 9 genes were ultimately selected (Appendix A).

### 2.3. Sample Characteristics and HWE Test

The subjects used in the present association study included 500 MSM with HIV-1 and 500 healthy controls. The basic characteristics of the patients with AIDS and healthy controls are summarized in Table 1. The mean age of the cases was 35.22 ± 11.81 years, and the mean age of the controls was 36.60 ± 14.32 years. There was no significant age difference between the two groups (*p* = 0.649). The genotypes of 37 candidate SNPs from 9 target genes of all subjects were identified. The genotyping results showed that none of the SNPs deviated from Hardy–Weinberg equilibrium (HWE) (*p* < 0.05) (Appendix A).

### 2.4. Association of 37 Candidate SNPs with HIV-1 Infection

We investigated the differences in the distribution of alleles and genotypes between patients and controls using the chi-square test to identify associations between SNPs and HIV-1 infection. The results showed that individuals carrying the G allele at *MAVS rs16989000* had a significantly higher probability of HIV-1 infection compared to those carrying the A allele (*p* = 0.049, OR = 1.197, 95% CI 1.001–1.433) (Appendix A).

Then, the effect of genotypes on HIV-1 infection was explored and significant differences in genotype distribution were observed (*p* < 0.05) (Appendix A). At *NLRP3 rs4612666*, individuals with the TT and TC genotypes had a higher risk of HIV-1 infection compared to those with the CC genotype (*p* = 0.040, OR = 1.325, 95% CI 1.012–1.733). Among the seven SNPs on *MAVS* gene screening, individuals with the GC genotype of *rs17857295* showed a higher risk of HIV-1 infection compared to those with the GG genotype (*p* = 0.034, OR = 1.392, 95% CI 1.025–1.890). Individuals with the TT and TC genotypes of *rs6084497* were associated with a higher risk of HIV-1 infection compared to those with the CC genotype (*p* = 0.046, OR = 1.293, 95% CI 1.004–1.665). Similarly, individuals with the CC and CA genotypes of *rs16989000* presented a higher risk of HIV-1 infection compared to those with the AA genotype (*p* = 0.041, OR = 1.309, 95% CI 1.011–1.696). Furthermore, individuals with the TG genotype at *JAK1 rs4244165* were associated with an increased risk of HIV-1 infection compared to those with the GG genotype (*p* = 0.019, OR = 1.373, 95% CI 1.054–1.788) (Figure 4).

### 2.5. Association of 37 Candidate SNPs with CD4^+^ T Cell Counts in People with HIV-1

CD4^+^ T lymphocytes are the main target cells of HIV-1 infection, and their count serves as a crucial indicator of HIV progression [18]. According to the latest National Health Commission of the People’s Republic of China WS-293-2019 standard for diagnosing AIDS [19], the samples with HIV-1 in this study were categorized into groups of 200 cells/μL and 500 cells/μL.

A chi-squared test was conducted to evaluate the association between alleles and CD4^+^ T cell counts in patients with HIV-1. The results showed that the *STAT1 rs2066804* G allele was significantly associated with a reduction in CD4^+^ T cells compared to the A allele in both the 200 cells/μL subgroup (*p* = 0.041, OR = 1.358, 95% CI 1.013–1.821) and the 500 cells/μL subgroup (*p* = 0.011, OR = 1.450, 95% CI 1.089–1.932). Additionally, the *IL1B rs1143623* C allele was significantly associated with CD4^+^ T cell reduction compared to the G allele in the 500 cells/μL subgroup (*p* = 0.009, OR = 1.465, 95% CI 1.098–1.955) (Appendix A).

The effect of genotype frequency distribution on CD4^+^ T cell counts in patients with HIV-1 was examined by the Wilcoxon rank-sum test. Individuals with the GG genotype at *STAT1 rs2066804* had lower CD4^+^ T cell counts compared to those with the AA genotype. Under the dominant model, individuals with the CC and GC genotypes at *STAT1 rs1467199* had lower CD4^+^ T cell counts compared to those with the GG genotype (Appendix A) (Figure 5).

### 2.6. Association of 37 Candidate SNPs with AIDS Staging in People with HIV-1

According to the clinical staging criteria of the World Health Organization (WHO) [20], AIDS cases were divided into two subgroups. The first subgroup, stages I and II of WHO criteria, corresponded to the acute and pre-AIDS stages as outlined in the Chinese Guidelines for the Diagnosis and Treatment of AIDS [21]. The second subgroup, stages III and IV, corresponded to the AIDS stage.

In this study, a chi-square test was used to evaluate significant differences between different stages to identify the relationship between genotypes and AIDS staging in patients (*p* < 0.05) (Appendix A). The analysis showed that individuals with the GA genotype of *OAS1 rs1131454* were associated with a later AIDS clinical stage compared to those with the AA genotype (*p* = 0.013, OR = 1.697, 95% CI 1.117–2.577). Similarly, individuals with the CC genotype at *NLRP3 rs10754558* showed a later AIDS clinical stage compared to those with the GC and GG genotypes (*p* = 0.027, OR = 1.532, 95% CI 1.048–2.239). Conversely, individuals with the AA genotype of *MAVS rs867335* were associated with a later clinical stage of AIDS compared to those with the AT genotype (*p* = 0.042, OR = 2.300, 95% CI 1.012–5.226) (Figure 6).

### 2.7. Interactions of Positively Associated SNPs and Their Combined Effect for AIDS

Generalized multifactor dimension reduction (GMDR; http://www.healthsystem.virginia.edu/internet/addiction-genomics/software/) (accessed on 9 July 2024) analysis was conducted to better understand the impact of candidate SNP interactions on HIV-1 infection and AIDS progression. Initially, we grouped people with HIV-1 and healthy controls for HIV-1 infection risk analysis. Among various locus models, the optimal combination was a five-locus model consisting of *rs17857295*, *rs4244165*, *rs6084497*, *rs16989000*, and *rs4612666* (test accuracy = 0.567, CVC = 10/10, *p* = 0.011) (Table 2 and Appendix A). Subsequently, we grouped patients with HIV-1 according to the stage of AIDS for disease progression assessment. The results indicated that the best combination was a three-locus model comprising *rs867335*, *rs10754558*, and *rs1131454* (test accuracy = 0.554, CVC = 10/10, *p* = 0.011) (Table 2 and Appendix A).

The cumulative effect of the optimal model was evaluated by the chi-square test to further study the impact of gene interaction on HIV-1 infection. When taking individuals without risk alleles as the reference group for comparison, the risk of HIV-1 infection in individuals with five risk alleles increased by 3.072 times (OR = 3.072, 95% CI 1.130–8.351, *p* = 0.022), the risk of individuals with seven risk alleles increased to 3.379 times (OR = 3.379, 95% CI 1.163–9.823, *p* = 0.021), and the risk of individuals with eight risk alleles increased to 4.000 times (OR = 4.000, 95% CI 1.052–15.207, *p* = 0.038) (Table 3). In addition, when examining broader categories, compared with individuals without risk alleles, individuals with 4–7 risk alleles had a 2.575-fold higher risk (OR = 2.575, 95% CI 0.976–6.792, *p* = 0.048). For individuals with 8–10 risk alleles, the risk further increased to 3.889 times (OR = 3.889, 95% CI 1.099–13.764, *p* = 0.032) (Figure 7). In conclusion, with the increase in the number of risk alleles carried by individuals, the risk of HIV-1 infection showed a significantly increasing trend. Similarly, taking individuals without risk genes as a reference, we found that the cumulative effect of gene interaction on the progression of AIDS was not significant (*p* > 0.05) (Appendix A).

## 3. Discussion

This study focused on the NLR signaling pathway closely related to AIDS through a series of bioinformatic analyses. Then, potential functional genetic variants were screened from the key genes of this pathway by functional annotation. Based on these findings, we conducted a case–control association study to explore the relationship between candidate SNPs and HIV-1 infection and AIDS progression. In addition, the established gene interaction model was used for positively associated SNPs to explore the cumulative effect on AIDS. The NLR signaling pathway is crucial for recognizing and clearing HIV-1. *NLRP3* can be activated through various pathways, such as the OAS/RNaseL system [22]. Upon activation, NLRP3 assembles with caspase-1 and ASC to form the NLRP3 inflammasome. This complex drives the maturation and secretion of IL-1β and IL-18, leading to inflammatory response and pyroptosis [14,15]. In the early stages of HIV-1 infection, IFN-α and IFN-β are produced. However, this initial response is usually inhibited by the viral escape mechanism [23].

This study demonstrates that *NLRP3* SNPs are associated with HIV-1 infection and AIDS progression. Previous studies have indicated that missense mutations in *NLRP3* impair the mouse immune system’s recruitment of IL-1β, reducing its ability to clear the influenza virus [24,25]. Our findings reveal that individuals with genotypes TT and TC at *NLRP3 rs4612666* are more susceptible to HIV-1 infection than those with the CC genotype. This is consistent with previous research, which indicated that the TT genotype was associated with an increased risk of tuberculosis infection and the T allele heightened the risk of extrapulmonary tuberculosis [26]. Furthermore, the TC genotype has been reported to be associated with a higher risk of infection with periodontitis [27]. The functional annotation from GWAS4D suggests that *rs4612666* may affect the transcription efficiency of immune-related genes (such as *CD59* and *STAT6*) by reducing the affinity of transcription factors or DNA-binding proteins for motif binding [28]. The above evidence suggests that *rs4612666* may impair the virus clearance ability of the human body, thereby increasing the risk of viral infectious diseases such as HIV-1. In addition, individuals with the CC genotype at *NLRP3 rs10754558* are associated with more advanced clinical stages. A previous study has showed that the C allele frequency was significantly higher in patients with HIV-1 compared with healthy controls in Italian and Brazilian populations [29]. Moreover, it has been reported that an increased frequency of the C allele in high-risk patients with HPV and cervical intraepithelial neoplasia is associated with adverse disease progression [30]. Although eQTL analysis shows that different genotypes of *rs10754558* do not significantly influence *NLRP3* expression level, a study has found that individuals with the CC genotype exhibit a lower *IL1B* expression level compared to those with the GG genotype [31].

This study shows that four *MAVS* SNPs affect HIV-1 infection and the progression of AIDS. Notably, individuals with the GC genotype at *MAVS rs17857295* present an elevated risk of HIV-1 infection compared to those with the CC genotype. Previous studies have shown that *rs17857295* C > G could significantly reduce the clearance rate of the hepatitis B virus (HBV) [32]. Functional annotations predicts that SNPs can alter binding specificity with transcription factors such as POLR2A, which is involved in the transcription of the HIV-1 genome and the formation of HIV-1 elongation complexes, as recorded in the Reactome database. This SNP may impact virus clearance rates, thereby influencing the risk of HIV-1 infection. Furthermore, individuals with the CC and CA genotypes at *MAVS rs16989000* have a higher risk of infection compared to those with the AA genotype. eQTL evidence shows that individuals with the CC genotype exhibit significantly lower *MAVS* expression levels than those with the CA and AA genotypes in peripheral blood. It has been reported that reduced *MAVS* expression can inhibit DDX3-MAVS signaling, thereby accelerating viral replication in people with HIV-1 [33]. Consequently, we infer that individuals with the CC genotype may experience an increased replication rate of HIV-1, leading to an increased risk of HIV-1 infection.

The positive results of the remaining genes also warrant further investigation. The C allele of *rs1143623*, located in the promoter region of *IL1B*, is significantly associated with decreased CD4^+^ T cell counts compared to the G allele. A previous study reported that individuals with CC and GC genotypes are associated with an increased risk of colorectal cancer compared to those with the GG genotype, and carrying the C allele is related to a higher disease risk [34]. Similar findings have been reported in patients with HBV and liver failure, where the frequency of the CC genotype is significantly higher than in healthy people [35]. Additionally, a study indicated that patients with colorectal cancer exhibited higher levels of *IL1B* compared to healthy controls, and individuals with the CC genotype demonstrated elevated mRNA levels compared to those with the GG genotype [36]. This is significant, as IL1B secretion has been reported to stimulate HIV-1 replication in T cells, potentially accelerating CD4^+^ T cell exhaustion [37]. These findings suggest that *rs1143623* may influence *IL1B* expression and affect CD4^+^ T cell counts in AIDS and related complications. In addition, our study identified that individuals with the GA genotype at *OAS1 rs1131454* were associated with a later clinical stage of AIDS compared to those with the AA genotype. eQTL evidence indicates that the *OAS1* expression level is higher in individuals with GA genotype than those with the AA genotype in blood samples. Previous research suggested that *OAS1* was significantly upregulated in people with AIDS with high viral loads compared to those with low viral loads [38], implying that increased *OAS1* expression may contribute to an excessive immune inflammatory response and the progression of AIDS.

In the in-depth exploration of the pathogenic role of positively associated SNPs, we categorized them into two main types: the type associated with HIV-1 infection and the type associated with AIDS progression. A further gene–gene interaction analysis showed that individuals with 4–7 risk alleles had a 2.575-fold increased risk of HIV-1 infection compared to individuals without risk alleles. The risk increased to 3.889-fold for individuals carrying 8–10 risk alleles. These findings indicate that the risk alleles have a cumulative effect on the risk of HIV-1 infection, because the risk of infection gradually increases with the increase in the number of risk alleles. It is worth noting that the current sample size included a very small number of individuals with 9 or 10 risk alleles, which limited the ability to ultimately determine the risk associated with these allele counts. This study is the first to report the cumulative effect of these specific risk alleles on the risk of HIV-1 infection. Future studies should increase the sample size to verify the results of gene interactions and conduct further investigations through cell and animal experiments.

The purpose of this study was to investigate the correlation between genetic variations in the NLR signaling pathway and HIV-1 infection and AIDS progression. Our findings are expected to provide theoretical support for optimizing precision medicine strategies in ART, promoting combination therapies, selecting vaccine antigens based on host genetic profiles, and identifying new molecular targets for therapeutic interventions. However, there are several limitations requiring improvement in this study. Firstly, the sample size needs to be increased to enhance the validity of the association results and re-duce the risk of Type I error. Secondly, to avoid potential population stratification bias, it is also crucial to conduct replication association studies on the relationship between genetic variations in the NLR signaling pathway and AIDS using different ethnic groups. Thirdly, it is extremely difficult to recruit MSM as a control group due to privacy concerns. Therefore, this study used a male healthy control group as an alternative, which may have introduced potential biases such as environmental and behavioral factors. Finally, the positively associated SNPs identified in this study warrant additional investigation through cell and animal experiments to provide a more comprehensive understanding of their implications.

## 4. Materials and Methods

### 4.1. Dataset Collection and Pre-Processing

Four datasets related to AIDS (GSE195434, GSE157198, GSE29429, and GSE2171) were obtained from the GEO database. These datasets were derived from whole blood or peripheral blood mononuclear cells (PBMCs). Patients with AIDS in the dataset who were treatment-naive constituted the case group for the following bioinformatic analysis, and the minimum sample size of both the case group and the control group was 10. To correct batch effects among samples, the R package “sva” was used to remove outliers.

### 4.2. Differential Expression Analysis and GSEA

For differential gene expression analysis, the “limma” package was used for expression profile by array data, while the “DESeq2” package was employed for the expression profile by high-throughput sequencing data. The screening criteria was |log_2_ FC| > 0.5. The R package “GSEA” was utilized for GSEA, and it was determined that the nominal *p* value (Nom *p* value) < 0.05 and the false discovery rate (FDR) < 0.25 were criteria for significant enrichment.

### 4.3. Cell-Type Deconvolution Analysis

The immune cell types in the GSE29429 dataset were obtained using the CIBERSORTx (https://cibersortx.stanford.edu/, accessed on 12 June 2023), with the LM22 feature matrix serving as a reference. The distribution of immune cells within the dataset was analyzed to compare differences between the case group and the control group through R 4.2.3.

### 4.4. WGCNA and KEGG Pathway Enrichment

WGCNA was employed to identify genes associated with important clinical indicators and construct gene modules. The Pearson correlation coefficient was used to express the strength of these correlations. The R package “WGCNA” was utilized to correlate the clinical information with the different gene modules, which were strongly related to the clinical indicators obtained under optimal threshold conditions. Subsequently, key genes within these modules underwent KEGG enrichment analysis. Pathways with FDR < 0.05 were selected for further investigation.

### 4.5. Selection of SNPs

After downloading SNPs corresponding to the gene segment from the 1000 Genomes Project (https://www.internationalgenome.org/), we performed functional annotation on these SNPs. The tools used for annotation included GWAS4D (http://www.mulinlab.org/gwas4d) (accessed on 10 October 2023), SNPinfo (https://snpinfo.niehs.nih.gov/) (accessed on 23 November 2023), Regulome DB (http://www.regulomedb.org/) (accessed on 10 October 2023), Reactome (https://reactome.org/) (accessed on 23 November 2023), GTEx (https://gtexportal.org/home/) (accessed on 10 October 2023), and Haploview v.4.2 (http://sourceforge.net/projects/haploview/) (accessed on 12 December 2023). Locus linkage disequilibrium (LD) analysis was conducted using Haploview v.4.2. SNPs with r^2^ > 0.8 were considered to be strongly linked. We identified functional SNPs through a comprehensive evaluation that included functional regions, mutation impact, expression quantitative trait loci (eQTL), and other potential pathogenic evidence. The principles for selecting SNPs in key pathway genes were as follows: (1) SNPs with blood eQTL signals in intronic regions were included if pathogenic evidence was present; (2) SNPs with blood eQTL signals in non-intronic gene regions were included; (3) SNPs without blood eQTL signals but supported by harmful predictions or pathogenic evidence from the literature were included; (4) SNPs in the promoter region with positive association evidence from the literature were included.

### 4.6. Sample Collection and Ethical Statement

In this study, 500 MSM with HIV-1 were recruited from the Centers for Disease Control and Prevention of Heilongjiang Province in Northern China as the case group. Additionally, 500 healthy men were recruited from the Second Affiliated Hospital of Harbin Medical University to serve as the control group. At the beginning of sample collection, we calculated the statistic power of the samples using Quanto software [39]. The MAF range of all candidate SNPs was from 0.24 to 0.45. If the false-positive rate of this association study was fixed at 0.05, the adopted sample size provided statistical power ranging from 80.1% to 89.5% for the detection of moderate-risk alleles with an odds ratio (OR) of 1.5. The study received approval from the Institutional Research Committee of Harbin Medical University. All procedures adhered to the principles outlined in the Declaration of Helsinki. Informed consent was obtained from all the participants.

### 4.7. Genotyping

In this study, the SNPscan^TM^ multiple SNP genotyping technology was used to genotype the functional SNPs. This technique relies on the high specificity of the ligase ligation reaction to identify SNPs. To obtain the ligation products corresponding to SNPs, non-specific sequences of varying lengths were introduced into the terminal segment of the ligation probe during the ligase addition reaction. These ligation products were then amplified using polymerase chain reaction (PCR) with universal primers labeled with fluorescence. A pre-mixed solution containing 2× PCR Master Mix (Genesky Biotechnologies Inc., Shanghai, China), Primer Mix (I or II) (Sangon Biotech Co., Ltd., Shanghai, China), ligation product, and ddH_2_O was prepared and supplemented with 1 μL ligation product. The PCR protocol involved four phases: (1) pre-denaturation at 95 °C for 2 min; (2) gradient amplification (denaturation at 94 °C for 20 s, annealing at 62 °C for 40, and extension at 72 °C 1.5 min for 9 cycles, with the annealing temperature decreasing by 0.5 °C reduction per cycle); (3) constant-temperature amplification (denaturation at 94 °C for 20, annealing at 57 °C for 40, and extension at 72 °C 1.5 min for 25 cycles); and (4) final extension at 68 °C for 60 min followed by 4 °C storage. The resulting products were separated by electrophoresis. Finally, the genotypes of each SNP were determined by analyzing the resulting electrophoretic map. To ensure quality control, genotyping was repeated for 5% of randomly selected case and control samples to verify the accuracy of the genotyping process. The repeatability of the genotyping results was confirmed to be 99.89%.

### 4.8. Statistical Analysis

Student’s *t*-test was used to compare the age distributions of the case and control groups. SNPs conforming to HWE were screened by Pearson’s chi-square test through calculating the genotype distributions between the two groups. The two-sided chi-square test was utilized to compare allele and genotype frequencies between the case group and the control group, as well as between groups with different clinical stages of AIDS. Yates’s continuity correction was applied when the expected frequencies were in the range of 1 to 5 (1 ≤ *T* < 5). The Wilcoxon rank-sum test was applied to compare CD4^+^ T lymphocyte counts across different genetic models. The genetic association analysis of genotypes with HIV-1 infection or AIDS involved three genetic models: the dominant model (homozygous risk + heterozygous vs. homozygous non-risk allele), the recessive model (homozygous risk vs. heterozygous + homozygous non-risk allele), and the codominant model (homozygous risk vs. heterozygous vs. homozygous non-risk allele). The optimal model for calculating gene–gene interactions was selected based on cross-validation consistency (CVC) ≥ 8/10 and test accuracy (TA) score ≥ 0.55, using the GMDR. The risk of candidate SNPs associated with HIV-1 infection and AIDS progression was measured using OR and 95% confidence intervals (CIs). A statistical analysis was conducted using SPSS version 26.0. The *p* value of 0.05 or less was considered statistically significant.

## 5. Conclusions

We found significant association of SNPs in NOD-like receptor signaling pathway genes with HIV-1 infection and AIDS progression. Specially, five SNPs (*rs17857295*, *rs4244165*, *rs6084497*, *rs16989000*, and *rs4612666*) were associated with HIV-1 infection, three SNPs (*rs2066804*, *rs1143623*, and *rs1467199*) were associated with CD4 T^+^ cell counts in patients with AIDS, and three SNPs (*rs867335*, *rs10754558*, and *rs1131454*) were found to be related to the clinical staging of AIDS.

## Figures and Tables

**Figure 1 ijms-26-03484-f001:**
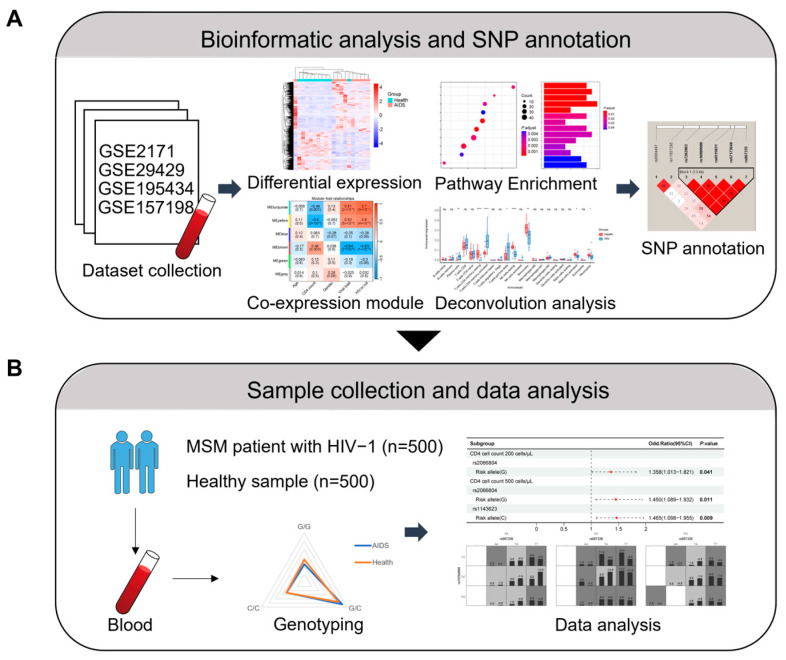
Schematic diagram of the study design. (**A**) Types of bioinformatic analysis performed. (**B**) Sample collection and genotyping data analysis. SNP, single-nucleotide polymorphism; MSM, men who have sex with men; and HIV-1, human immunodeficiency virus 1.

**Figure 2 ijms-26-03484-f002:**
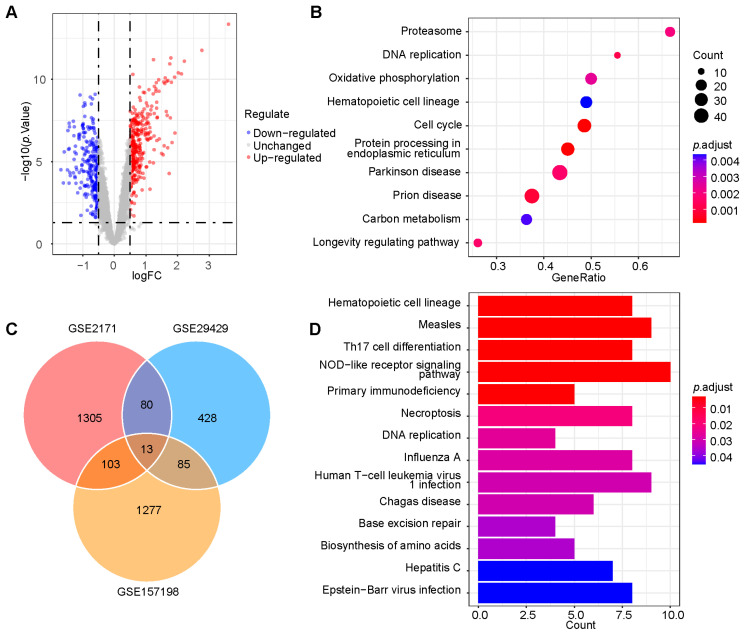
Screening of differentially expressed genes and pathway enrichment. (**A**) Volcano plot of 606 most differentially expressed genes in GSE29429. (**B**) GSEA for differentially expressed genes in GSE29429. (**C**) Intersection of differential expression analysis results of datasets GSE29429, GSE2171, and GSE157198. (**D**) KEGG enrichment analysis on the intersection results. AIDS, acquired immunodeficiency syndrome; GSEA, gene set enrichment analysis.

**Figure 3 ijms-26-03484-f003:**
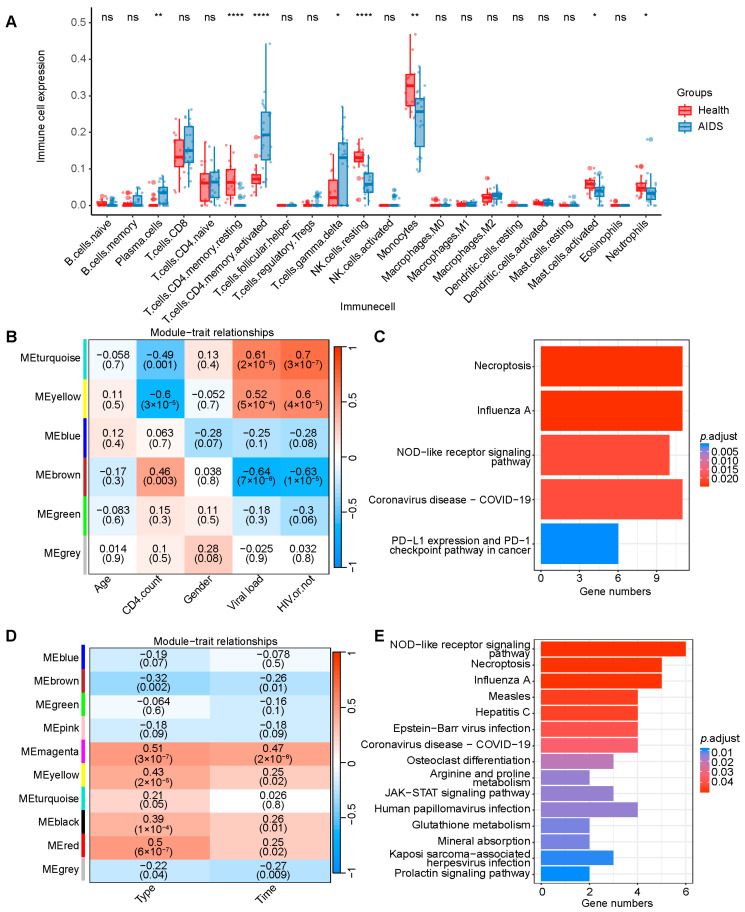
Cell-type deconvolution analysis and WGCNA. (**A**) The different composition of immune cells in GSE29429 between patients with AIDS and healthy controls. ns, no significance; Wilcoxon rank-sum test: * *p* ≤ 0.05, ** *p* < 0.01, and **** *p* < 0.0001. (**B**) Correlations between gene module and clinical features, each module containing the corresponding correlation coefficient and *p* value. The “MEyellow” module has the highest correlation with clinical features in GSE29429. (**C**) Enrichment results of KEGG pathway in “MEyellow” module. (**D**) Correlations between gene module and clinical features. The “MEmagenta” module has the highest correlation with clinical features in GSE157198. (**E**) Enrichment results of KEGG pathway in “MEmagenta” gene module. AIDS, acquired immunodeficiency syndrome; WGCNA, weighted gene co-expression network analysis; and KEGG, Kyoto Encyclopedia of Genes and Genomes.

**Figure 4 ijms-26-03484-f004:**
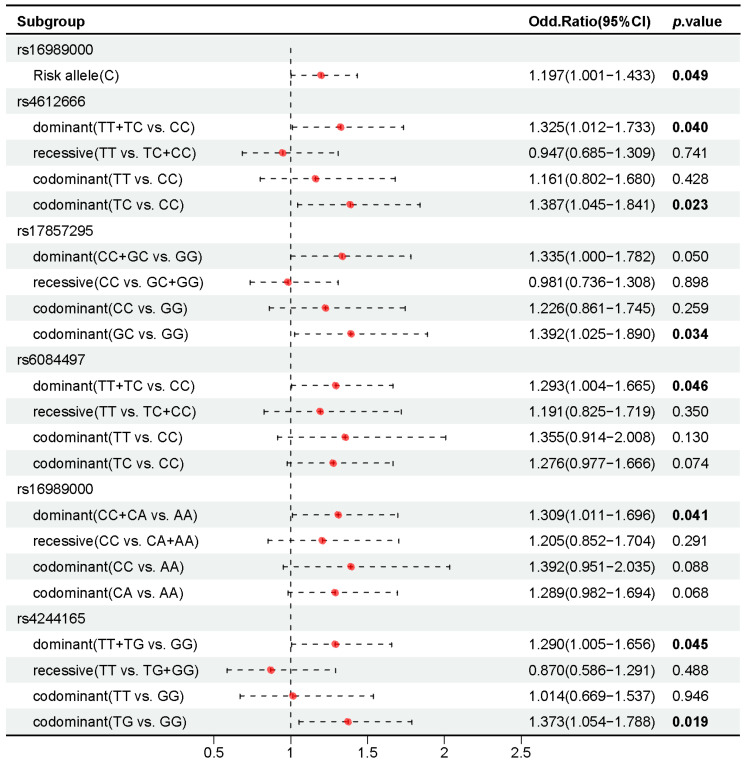
Association of alleles and genotypes of some candidate SNPs with HIV-1 infection. Bold type indicates statistical significance (*p* < 0.05). CI, confidence interval.

**Figure 5 ijms-26-03484-f005:**
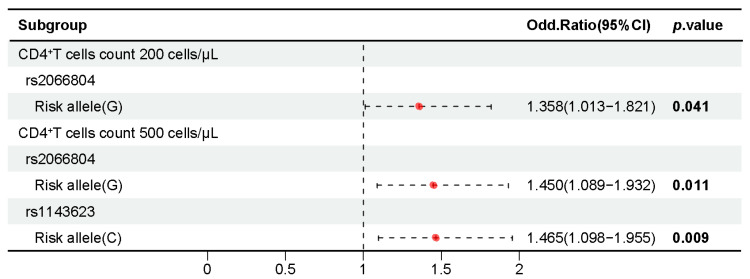
Association of alleles and genotypes of some candidate SNPs with CD4^+^ T cell counts in people with HIV-1. Bold type indicates statistical significance (*p* < 0.05). CI, confidence interval.

**Figure 6 ijms-26-03484-f006:**
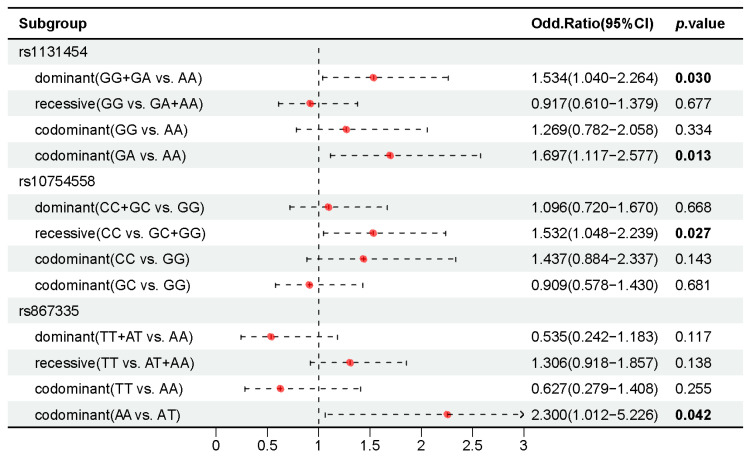
Association of some candidate SNPs with clinical stage of patients with AIDS. Bold type indicates statistical significance (*p* < 0.05). CI, confidence interval.

**Figure 7 ijms-26-03484-f007:**
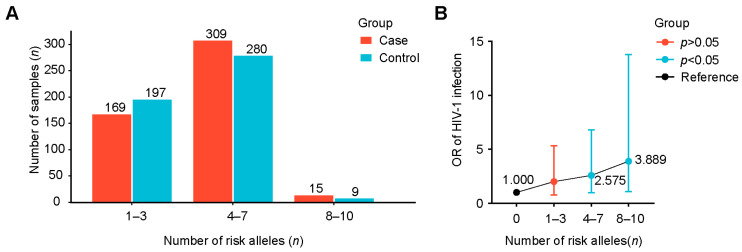
Association between HIV-1 infection and the number of risk alleles in different groups. (**A**) The number of samples in different risk alleles. (**B**) Association between risk allele groups and HIV-1 infection, taking individuals without risk alleles as a reference. OR, odds ratio; CI, confidence interval.

**Table 1 ijms-26-03484-t001:** Basic characteristics of cases and controls.

Variable	Case (*n* = 500)	Control (*n* = 500)	*p* Value
Age range (years)	17–81	16–78	-
Mean age ± SD, years	35.22 ± 11.81	36.60 ± 14.32	0.649 ^a^
Gender *n* (%)			
Male	500 (100%)	500 (100%)	-
Female	-	-	-
Clinical stages, *n* (frequency)			
Ⅰ	125 (25.0%)	-	-
Ⅱ	125 (25.0%)	-	-
Ⅲ	190 (38.0%)	-	-
Ⅳ	60 (12.0%)	-	-
CD4^+^ T cell counts (cells/μL), *n* (frequency)			
<200	124 (24.8%)	-	-
200–349	126 (25.2%)	-	-
350–500	125 (25.0%)	-	-
≥500	125 (25.2%)	-	-

^a^ Student’s *t*-test. SD, standard deviation.

**Table 2 ijms-26-03484-t002:** GMDR analysis for the best gene–gene interaction models.

Best Combination	Training Accuracy	Testing Accuracy	CV Consistency	Sign Test (*p*)
Infection module				
*rs4244165*	0.540	0.508	7/10	4(0.828)
*rs4244165 rs6084497*	0.555	0.477	4/10	2(0.989)
*rs4244165 rs6084497 rs4612666*	0.581	0.525	5/10	6(0.377)
*rs17857295 rs4244165 rs6084497 rs4612666*	0.620	0.520	7/10	6(0.377)
*rs17857295 rs4244165 rs6084497 rs16989000 rs4612666*	0.670	0.567	10/10	9(**0.011**)
Stage module				
*rs1131454*	0.556	0.494	5/10	5(0.623)
*rs867335 rs1131454*	0.577	0.493	7/10	4(0.828)
*rs867335 rs10754558 rs1131454*	0.605	0.554	10/10	**9(0.011)**

Bold type indicates statistical significance (*p* < 0.05).

**Table 3 ijms-26-03484-t003:** Association between the number of risk alleles and HIV-1 infection.

Risk Allele (*n*)	Case with Risk Allele	Case Without Risk Allele	Control with Risk Allele	Control Without Risk Allele	*p* Value	OR (95%CI)
1	23	6	35	14	0.441	1.533 (0.515–4.567)
2	67	6	64	14	0.078	2.443 (0.884–6.746)
3	79	6	98	14	0.210	1.881 (0.691–5.119)
4	108	6	111	14	0.098	2.270 (0.842–6.124)
5	104	6	79	14	**0.022**	3.072 (1.130–8.351)
6	55	6	61	14	0.148	2.104 (0.756–5.854)
7	42	6	29	14	**0.021**	3.379 (1.163–9.823)
8	12	6	7	14	**0.038**	4.000 (1.052–15.207)
9	3	6	2	14	0.312	3.500 (0.460–26.616)
10	0	6	0	14	-	-

Bold type indicates statistical significance (*p* < 0.05).

## Data Availability

The original contributions presented in this study are included in the article/Appendix A; further inquiries can be directed to the corresponding author.

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
