# Peer review of "Genetic Variants in the NOD-like Receptor Signaling Pathway Are Associated with HIV-1/AIDS in a Northern Chinese Population"

_ijms, 2025, doi:10.3390/ijms26083484_

Round 1

Reviewer 1 Report

Comments and Suggestions for Authors The authors performed an association analysis using SNPs in genes of the NLR signaling pathway that have been described to influence HIV infection and the progression of immunodeficiency syndrome (AIDS). Using a series of bioinformatics analyses and functional annotations, the authors selected SNPs in 9 genes of the NLR signaling pathway.   Several concerns are raised in the selection of SNPs and in the association analyses. Mainly in the description of the population and the phenotype.   They are presented in the following points:   1.Why did the authors not analyze SNPs in the 13 genes obtained from the intersection of the results of the differential expression analysis of the data sets GSE29429, GSE2171 and GSE157198?   2.In section 4.5. SNP selection, describe the criteria by which a SNP was selected as a functional SNP included in table s1. Why were only 9 genes ultimately genotyped?   3.Please describe the genotyping method in more detail. What is the established name of the method used for genotyping? Alternatively, use a reference to define the method   4.What multiple testing correction was used? Bonferroni, FDR in the association analysis?   5.Men included in the control group should have the same characteristics and share the same environment as HIV-positive MSM. It is mandatory that exposure to HIV is similar between groups. Authors should indicate that the control group was exposed to HIV infection like HIV-positive MSM.   6.On the other hand, authors should indicate that the HIV+ group is naïve and not treated with ART. In addition, the time since HIV infection should be included in Table 1 to ensure that this parameter is the same across all clinical stages and CD4 cell counts. Both ART and the time of infection affect CD4 count and the development of AIDS.   7.The first paragraph of the discussion should summarize the methods and the main results obtained.            

Reviewer 2 Report

Comments and Suggestions for Authors

Genetic variants in the Nod-like receptor signaling pathway are associated with HIV-1/AIDS in the Northern Chinese MSM population.

This study investigates the genetic association between single nucleotide polymorphisms (SNPs) in the NOD-like receptor (NLR) signaling pathway and HIV-1/AIDS progression in men who have sex with men (MSM) from Northern China. The NLR pathway is crucial in immune response regulation, and its genetic variants may influence HIV susceptibility and disease progression.

Using bioinformatics analysis and functional annotation, the study identified 37 SNPs in key NLR pathway genes. A case-control study was conducted on 500 HIV-1-positive MSM and 500 healthy controls, revealing that five SNPs (NLRP3 rs4612666, MAVS rs17857295, MAVS rs6084497, MAVS rs16989000, and JAK1 rs4244165) were significantly associated with HIV-1 infection. Additionally, three SNPs (IL1B rs1143623, STAT1 rs1467199, and STAT1 rs2066804) correlated with CD4+ T cell counts, and three others (OAS1 rs113145, NLRP3 rs1075455, and MAVS rs867335) were linked to AIDS clinical staging.

Line 23-28:  The phrase "can influence human immunodeficiency virus (HIV) clearance and CD4+ T cell recovery" is somewhat strong—consider specifying "may influence" unless supported by substantial experimental evidence.

Line 41-48: "Recent studies indicate that innate immunity limits viral growth and slows CD4+T cell depletion when ART is discontinued" – include a references.

Lines 75-106: Three datasets (GSE157198, GSE29429, and GSE2171) related to HIV infection were selected from the GEO database." – Briefly explain why these datasets were chosen over others (e.g., sample size, RNA-seq vs. microarray, relevance).

Lines 125-130: SNP selection from 1000 Genomes Project. How were linkage disequilibrium (LD) blocks determined? Mention the r² threshold.

Lines 347-354: Sample collection and ethics:  Why were only men who have sex with men (MSM) included? Briefly justify the choice of this population (higher HIV risk, genetic homogeneity, etc.).

Lines 132-138: Participant demographics. Why were only men who have sex with men (MSM) included? Briefly justify the choice of this population (higher HIV risk, genetic homogeneity, etc.).

Lines 172-193: CD4+ T cell associations: The dominant and recessive models should be explicitly mentioned in the methods section before presenting results.

Lines 194-211: Association of SNPs with AIDS clinical staging: The significance threshold (p < 0.05) is mentioned, but given multiple SNP comparisons, discuss potential false positives due to multiple testing.

Lines 232-275: Interpretation of results.: "NLRP3 SNPs significantly influence both HIV-1 infection and AIDS progression" – soften this to "NLRP3 SNPs are associated with..." since causation is not established.

"Functional annotation from GWAS4D suggests that rs4612666 may decrease the motif binding affinity of immune-related genes, such as CD59 and STAT6." – Explain how this could affect immune response.

Lines 299-306: Limitations of the study. Mention lack of replication in an independent cohort. Consider discussing the potential role of environmental or behavioral factors influencing genetic susceptibility.

Potential population stratification bias (e.g., only Northern Chinese Han MSM).

Comments on the Quality of English Language

Genetic variants in the Nod-like receptor signaling pathway are associated with HIV-1/AIDS in the Northern Chinese MSM population.

This study investigates the genetic association between single nucleotide polymorphisms (SNPs) in the NOD-like receptor (NLR) signaling pathway and HIV-1/AIDS progression in men who have sex with men (MSM) from Northern China. The NLR pathway is crucial in immune response regulation, and its genetic variants may influence HIV susceptibility and disease progression.

Using bioinformatics analysis and functional annotation, the study identified 37 SNPs in key NLR pathway genes. A case-control study was conducted on 500 HIV-1-positive MSM and 500 healthy controls, revealing that five SNPs (NLRP3 rs4612666, MAVS rs17857295, MAVS rs6084497, MAVS rs16989000, and JAK1 rs4244165) were significantly associated with HIV-1 infection. Additionally, three SNPs (IL1B rs1143623, STAT1 rs1467199, and STAT1 rs2066804) correlated with CD4+ T cell counts, and three others (OAS1 rs113145, NLRP3 rs1075455, and MAVS rs867335) were linked to AIDS clinical staging.

Line 23-28:  The phrase "can influence human immunodeficiency virus (HIV) clearance and CD4+ T cell recovery" is somewhat strong—consider specifying "may influence" unless supported by substantial experimental evidence.

Line 41-48: "Recent studies indicate that innate immunity limits viral growth and slows CD4+T cell depletion when ART is discontinued" – include a references.

Lines 75-106: Three datasets (GSE157198, GSE29429, and GSE2171) related to HIV infection were selected from the GEO database." – Briefly explain why these datasets were chosen over others (e.g., sample size, RNA-seq vs. microarray, relevance).

Lines 125-130: SNP selection from 1000 Genomes Project. How were linkage disequilibrium (LD) blocks determined? Mention the r² threshold.

Lines 347-354: Sample collection and ethics:  Why were only men who have sex with men (MSM) included? Briefly justify the choice of this population (higher HIV risk, genetic homogeneity, etc.).

Lines 132-138: Participant demographics. Why were only men who have sex with men (MSM) included? Briefly justify the choice of this population (higher HIV risk, genetic homogeneity, etc.).

Lines 172-193: CD4+ T cell associations: The dominant and recessive models should be explicitly mentioned in the methods section before presenting results.

Lines 194-211: Association of SNPs with AIDS clinical staging: The significance threshold (p < 0.05) is mentioned, but given multiple SNP comparisons, discuss potential false positives due to multiple testing.

Lines 232-275: Interpretation of results.: "NLRP3 SNPs significantly influence both HIV-1 infection and AIDS progression" – soften this to "NLRP3 SNPs are associated with..." since causation is not established.

"Functional annotation from GWAS4D suggests that rs4612666 may decrease the motif binding affinity of immune-related genes, such as CD59 and STAT6." – Explain how this could affect immune response.

Lines 299-306: Limitations of the study. Mention lack of replication in an independent cohort. Consider discussing the potential role of environmental or behavioral factors influencing genetic susceptibility.

Potential population stratification bias (e.g., only Northern Chinese Han MSM).

Reviewer 3 Report

Comments and Suggestions for Authors

Genetic variants in the Nod-like receptor signalling pathway are 2 associated with HIV-1/AIDS in the Northern Chinese MSM 3 population. This study uses multiple bioinformatics tools and functional annotations to identify the single nucleotide polymorphism in NLR signalling pathways and how it influences HIV infection and disease progression

  1. In the title, on line 3. Please avoid using abbreviations in the title. It would be useful to say an at-risk population in place of the MSM abbreviation.
  2. In the introduction, HIV statistics must be updated. It would be good if the authors could highlight the role of ART and the problem statement that we are still facing in HIV research. What are the limitations of ART? Why the current study is important?
  3. In the second paragraph of your introduction, I recommend that you briefly describe how HIV affect the immune system. Which cells of the innate immune system are affected by HIV?
  4. On line 52, “The NLR signalling pathway is involved 52 in regulating inflammatory and immune responses during HIV-1 infection”. It might be beneficial to the reader to briefly say how the NLR signalling pathway regulates inflammatory and immune responses during HIV-1 infection.
  5. On line 78, “GEO” appear for the first time. Please write in full at first appearance and continue with abbreviations where appropriate.
  6. On line 80, “GSEA” appear for the first time. Please write in full at first appearance and continue with abbreviations where appropriate.
  7. On line 84, “KEGG” appear for the first time. Please write in full at first appearance and continue with abbreviations where appropriate.
  8. On lines 81-84, “In 81 order to comprehensively explore the influence of different expression genes”. Please rephrase this. I think the authors mean to say the expression of different genes.
  9. Several abbreviations are used for the first time in the text and are not said in full. It is important not to assume that the reader will know these abbreviations. Genes are written in italics, I suggest that all the SNPs be written in Italics
  10. How did you choose the 37 SNPs? In your analysis of HIV-positive individuals, were you able to classify between the chronic and acute infected individuals?
  11. Figure 3A, the Y-axis “Immunecell” is spelled incorrectly. It should be an immune cell.
  12. Figure s3 is not visible; I suggest that authors work on the resolution of the figure.
  13. The module-trait relationships provide interesting insights in Figures 3 B and D. However, it is not immediately apparent what genes or data is in these modules. It would be helpful for the reader to know what is included in the gene modules used, ideally detailed in the materials and methods section or the supplementary.
  14. In line 254, the author mentions the previous studies but only references one. Reword or add the other studies that you are referring to.
  15. Genotyping in section 4.7 is not well explained; may the authors provide the names of the kits used for this experiment and the PCR conditions? This section needs improvement so that it can be followed by the reader.
  16. The findings of this study indicated that the various genotypes studied influence the progression to AIDS, the pathogenicity of HIV, CD4+ T cell count, as well as other important biological processes and outcomes. Important comparisons were made to the findings of other studies in the discussion. However, what are the implications of these findings/insights, and how can they influence future research? This has not been answered in the discussion or conclusion despite the study's limitations outlined.  
  17. The study includes exclusively individuals from the Han population in China. A comparison of the major findings to other key demographics affected by HIV with regards to the NLR signaling pathway in the discussion would be beneficial if there are significant differences or similarities if relevant studies exist.

General comments: The undertaking of this study demonstrates an advanced understanding of bioinformatics and statistical analysis. The study at times assumes the reader is familiar with the tools, databases and technical terms used, it would be beneficial to explain the choices of databases, tools or analyses particularly in the methods where appropriate.

Round 2

Reviewer 1 Report

Comments and Suggestions for Authors

Many thanks to the authors for responding to the comments and making the suggested changes.

Author Response

Comments and Suggestions:

Many thanks to the authors for responding to the comments and making the suggested changes.

Response: We are very grateful indeed for the work you have done for improving the quality of our manuscript.

Reviewer 3 Report

Comments and Suggestions for Authors

Minor comments

The authors must check the use of pnctuation marks throughout the manuscript. And also double check whether all the genes are written in Italics

The authors has addressed all the comments.

Author Response

Comments and Suggestions:

Minor comments

The authors must check the use of punctuation marks throughout the manuscript. And also, double check whether all the genes are written in Italics

The authors have addressed all the comments.

Response: Thank you for the reviewer 's suggestion. We have made necessary changes to the format and grammar. The corresponding modifications in the manuscript are marked in red.